# Clustering with Deep Learning: Taxonomy and New Methods

## Abstract

Clustering is a fundamental machine learning method. The quality of its results is dependent on the data distribution. For this reason, deep neural networks can be used for learning better representations of the data. In this paper, we propose a systematic taxonomy for clustering with deep learning, in addition to a review of methods from the field. Based on our taxonomy, creating new methods is more straightforward. We also propose a new approach which is built on the taxonomy and surpasses some of the limitations of some previous work. Our experimental evaluation on image datasets shows that the method approaches state-of-the-art clustering quality, and performs better in some cases.

## 1 Introduction

Clustering is one of the most fundamental unsupervised machine learning problems. Its main goal is to separate data into clusters of similar data points. Besides having its own applications, it is beneficial for multiple other fundamental tasks. For instance, it can serve for automatic data labeling for supervised learning and as a pre-processing step for data visualization and analysis.

However, the performance of clustering algorithms is dependent on the type of the input data, such that different problems and datasets could require different similarity measures and different separation techniques. As a result, dimensionality reduction and representation learning have been extensively used alongside clustering, in order to map the input data into a feature space where separation is easier with respect to the problem's context. Using deep neural networks (DNNs), it is possible to learn non-linear mappings allowing to transform the data into more clustering-friendly representations.

In the past, dimensionality reduction (or representation learning) and clustering have been treated separately, and sequentially applied on the data (Ding and He, 2004; Tian et al., 2014; Trigeorgis et al., 2014). However, recent research has shown that jointly optimizing for both problems can achieve decent results (Song et al., 2013; Xie et al., 2016; Yang et al., 2016a;b; Li et al., 2017).

One of our main contributions is the formulation of a taxonomy of methods that use deep learning for clustering. Our taxonomy facilitates the overview of existing methods and the creation of new ones by using the best properties of the existing ones in a modular manner.

Based on the taxonomy, we propose a new method that combines advantageous properties of some existing methods. We use an autoencoder-based method for learning better representations of the data which are clustering-friendly, with a state-of-the-art training procedure. The training has two phases, the first one being standard autoencoder training with the mean squared error reconstruction loss, and the second one is based on a loss function combining the reconstruction loss and a clustering-specific loss. Moreover, in the second phase, we alternate between optimizing the network model, and updating the clustering assignments.

The rest of the paper is organized as follows: the taxonomy of clustering with deep learning and the corresponding building blocks is described in Section 2. In Section 3, several related methods are briefly described and compared based on the taxonomy. Subsequently, in Section 4, a new method is proposed and discussed based on the building blocks of the taxonomy. Results of the proposed method are shown in Section 5, followed by conclusions in Section 6.

## 2 TAXONOMY

The most successful methods for clustering with deep neural networks all work following the same principle: representation learning using DNNs and using these representations as input for a specific clustering method. Every method consists of the following parts, for each of which there are several options to choose from:

- Neural network training procedure, consisting of the following:
    - Main neural network branch and its usage
        * Architecture of main neural network branch, described in Section 2.1
        * Set of deep features used for clustering, described in Section 2.2
    - Neural network losses:
        * Non-clustering loss, described in Section 2.3
        * Clustering loss, described in Section 2.4
        * Method to combine the two losses, described in Section 2.5
    - Cluster updates, described in Section 2.6
- After the network training: re-run clustering (optional), described in Section 2.7

### 2.1 ARCHITECTURE OF MAIN NEURAL NETWORK BRANCH

In most deep learning methods for clustering, the "main branch" of the neural network (apart from side branches towards non-clustering losses, see Section 2.3) is used to transform the inputs into a latent representation that is used for clustering. The following neural network architectures have previously been used for this purpose:

- **Multilayer perceptron (MLP)**: Feedforward network, consisting of several layers of neurons, such that the output of every hidden layer is the input to next one.
- **Convolutional neural network (CNN)**: Inspired by biology, more precisely by the organization of the animal visual cortex. Useful for applications to regular-grid data such as images, if locality and shift-equivariance/invariance of feature extraction is desired.
- **Deep belief network (DBN)**: Generative graphical model, consisting of several layers of latent variables. It is composed of several shallow networks such as restricted Boltzmann machines, such that the hidden layer of each sub-network serves as the visible layer of the next sub-network.

### 2.2 SET OF DEEP FEATURES USED FOR CLUSTERING

DNNs serve for clustering as mappings to better representations. The features of these representations can be drawn from different layers of the network or even from several ones. It is possible to separate this choice into two categories:

- **One layer**: Refers to the general case where only the output of the last layer of the network is used. This approach benefits from the low dimensionality of the representation.
- **Several layers**: Refers to the case where the representation is a combination of the outputs of several layers. Based on that, the representation is richer and allows the embedded space to represent more complex semantic representations, which might enhance the separation process and help in the similarity computation (Saito and Tan, 2017).

### 2.3 NON-CLUSTERING LOSS

The non-clustering loss is independent of the clustering algorithm and usually enforces a desired constraint on the learned model. The following are possible options for non-clustering loss functions:

- **No non-clustering loss**: No additional non-clustering loss functions are used. In such cases, the network model is only constrained by the clustering loss requirements. For most

clustering losses, no non-clustering loss can have a danger of worse representations/results, or theoretically even collapsing clusters (Yang et al., 2016a), but the latter rarely occurs in practice.

- **Autoencoder reconstruction loss**: The autoencoder consists of two parts: an encoder and a decoder. The encoder maps its input $x$ to a representation $z$ in a latent space $Z$. During training, the decoder tries to reconstruct $x$ from $z$, making sure that useful information has not been lost by the encoding phase. In the context of clustering methods, once the training is done the decoder part is no longer used, and the encoder is left for mapping its input to the latent space $Z$. By applying this procedure, autoencoders can successfully learn useful representations in the cases where the output's dimensionality is different from the input's or when random noise is injected to the input (Vincent et al., 2010). Additionally, they can also be used for dimensionality reduction goals (Hinton and Salakhutdinov, 2006). Generally the reconstruction loss is a distance measure $d_{\mathrm{AE}}(x_i, f(x_i))$ between the input $x_i$ to the autoencoder and the corresponding reconstruction $f(x_i)$. One particular formulation of it is using the mean squared error of the two variables:

$$L = d_{\mathrm{AE}}(x_i, f(x_i)) = \sum_i \|x_i - f(x_i)\|^2, \tag{1}$$

where $x_i$ is the input and $f(x_i)$ is the autoencoder reconstruction. This loss function guarantees that the learned representation preserves important information from the initial one, which is why reconstruction is possible.

- **Other tasks**: Additional information about training samples that is available in the form of targets, even if not perfectly suitable to dictate clustering, can be used in a (multi-task) non-clustering loss to encourage meaningful feature extraction.

## 2.4 CLUSTERING LOSS

The second type of functions is specific to the clustering method and the clustering-friendliness of the learned representations, therefore such functions are called clustering loss functions. The following are options for clustering loss functions:

- **No clustering loss**: Even if a neural network has only non-clustering losses (Section 2.3), the features it extracts can be used for clustering after training (Sections 2.6–2.7). The neural network serves in this case for changing the representation of the input, for instance changing its dimensionality. Such a transformation could be beneficial for the clustering sometimes, but using a clustering loss usually yields better results (Xie et al., 2016; Yang et al., 2016a).

- **k-Means loss**: Assures that the new representation is $k$-means-friendly (Yang et al., 2016a), i.e. data points are evenly distributed around the cluster centers. In order to obtain such a distribution a neural network is trained with the following loss function:

$$L(\theta) = \sum_{i=1}^{N} \sum_{k=1}^{K} s_{ik} \|z_i - \mu_k\|^2, \tag{2}$$

where $z_i$ is an embedded data point, $\mu_k$ is a cluster center and $s_{ik}$ is a boolean variable for assigning $z_i$ with $\mu_k$. Minimizing this loss with respect to the network parameters assures that the distance between each data point and its assigned cluster center is small. Having that, applying $k$-means would result in better clustering quality.

- **Cluster assignment hardening**: Requires using soft assignments of data points to clusters. For instance, Student's $t$-distribution can be used as the kernel to measure the similarity (van der Maaten and Hinton, 2008) between points and centroids. This distribution $Q$ is formulated as follows:

$$q_{ij} = \frac{(1 + \|z_i - \mu_j\|^2/\nu)^{-\frac{\nu+1}{2}}}{\sum_{j'}(1 + \|z_i - \mu_{j'}\|^2/\nu)^{-\frac{\nu+1}{2}}}, \tag{3}$$

where $z_i$ is an embedded data point, $\mu_j$ is the $j^{th}$ cluster centroid, and $\nu$ is a constant, e.g. $\nu = 1$. These normalized similarities between points and centroids can be considered

as soft cluster assignments. The *cluster assignment hardening* loss then enforces making these soft assignment probabilities stricter. It does so by letting cluster assignment probability distribution $Q$ approach an auxiliary (target) distribution $P$ which guarantees this constraint. Xie et al. (2016) propose the following auxiliary distribution:

$$p_{ij} = \frac{q_{ij}^2/\Sigma_i q_{ij}}{\Sigma_{j'}(q_{ij'}^2/\Sigma_i q_{ij'})}. \tag{4}$$

By squaring the original distribution and then normalizing it, the auxiliary distribution $P$ forces assignments to have stricter probabilities (closer to 0 and 1). It aims to improve cluster purity, put emphasis on data points assigned with high confidence and to prevent large clusters from distorting the hidden feature space (Xie et al., 2016). One way to formulate the divergence between the two probability distributions is using the Kullback–Leibler divergence (Kullback and Leibler, 1951). It is formulated as follows:

$$L = \text{KL}(P\|Q) = \sum_i \sum_j p_{ij} \log \frac{p_{ij}}{q_{ij}}, \tag{5}$$

which is minimized for the aforementioned $Q$ and $P$ via neural network training.

- **Balanced assignments loss**: This loss has been used alongside other losses such as the previous one (Dizaji et al., 2017). Its goal is to enforce having balanced cluster assignments. It is formulated as follows:

$$L_{ba} = \text{KL}(G\|U) \tag{6}$$

where $U$ is the uniform distribution and $G$ is the probability distribution of assigning a point to each cluster:

$$g_k = P(y = k) = \frac{1}{N} \sum_i q_{ik} \tag{7}$$

By minimizing equation 6, the probability of assigning each data point to a certain cluster is uniform across all possible clusters (Dizaji et al., 2017). It is important to note that this property (uniform assignment) is not always desired. Thus, in case any prior is known it is still possible to replace the uniform distribution by the known prior one.

- **Locality-preserving loss**: This loss aims to preserve the locality of the clusters by pushing nearby data points together (Huang et al., 2014). Mathematically, it is formulated as follows:

$$L_{lp} = \sum_i \sum_{j \in N_k(i)} s(x_i, x_j)\|z_i - z_j\|^2 \tag{8}$$

where $N_k(i)$ is the set of $k$ nearest neighbors of the data point $x_i$, and $s(x_i, x_j)$ is a similarity measure between the points $x_i$ and $x_j$.

- **Group sparsity loss**: It is inspired by spectral clustering where block diagonal similarity matrix is exploited for representation learning (Ng et al., 2002). Group sparsity is itself an effective feature selection method. In Huang et al. (2014), the hidden units were divided into $G$ groups, where $G$ is the assumed number of clusters. When given a data point $x_i$ the obtained representation has the form $\{\phi^g(x_i)\}_{g=1}^G$. Thus the loss can be defined as follows:

$$L_{gs} = \sum_{i=1}^N \sum_{g=1}^G \lambda_g \|\phi^g(x_i)\|, \tag{9}$$

where $\{\lambda_g\}_{g=1}^G$ are the weights to sparsity groups, defined as

$$\lambda_g = \lambda\sqrt{n_g}, \tag{10}$$

where $n_g$ is the group size and $\lambda$ is a constant.

- **Cluster classification loss**: Cluster assignments obtained during cluster updates (Section 2.6) can be used as "mock" class labels for a classification loss in an additional network branch, in order to encourage meaningful feature extraction in all network layers (Hsu and Lin, 2017).

- **Agglomerative clustering loss**: Agglomerative clustering merges two clusters with maximum affinity (or similarity) in each step until some stopping criterion is fulfilled. A neural network loss inspired by agglomerative clustering (Yang et al., 2016b) is computed in several steps. First, the cluster update step (Section 2.6) merges several pairs [correct?] of clusters by selecting the pairs with the best affinity (some predefined measure of similarity between clusters). Then network training retrospectively even further optimizes the affinity of the already merged clusters (it can do so because the affinity is measured in the latent space to which the network maps). After the next cluster update step, the network training switches to retrospectively optimizing the affinity of the newest set of newly merged cluster pairs. In this way, cluster merging and retrospective latent space adjustments go hand in hand. Optimizing the network parameters with this loss function would result in a clustering space more suitable for (agglomerative) clustering.

## 2.5 METHOD TO COMBINE THE LOSSES

In the case where a clustering and a non-clustering loss function are used, they are combined as follows:

$$L(\theta) = \alpha L_c(\theta) + (1 - \alpha)L_n(\theta), \tag{11}$$

where $L_c(\theta)$ is the clustering loss, $L_n(\theta)$ is the non-clustering loss, and $\alpha \in [0; 1]$ is a constant specifying the weighting between both functions. It is an additional hyperparameter for the network training. It can also be changed during training following some schedule. The following are methods to assign and schedule the values of $\alpha$:

- **Pre-training, fine-tuning**: First, $\alpha$ is set to 0, i.e. the network is trained using the non-clustering loss only. Subsequently, $\alpha$ is set to 1, i.e. the non-clustering network branches (e.g. autoencoder's decoder) are removed and the clustering loss is used to train (fine-tune) the obtained network. The constraint forced by the reconstruction loss could be lost after training the network long enough for clustering only. In some cases, losing such constraints may lead to worse results (see Table 1).

- **Joint training**: $0 < \alpha < 1$, for example $\alpha = 0.5$, i.e. the network training is affected by both loss functions.

- **Variable schedule**: $\alpha$ is varied during the training dependent on a chosen schedule. For instance, start with a low value for $\alpha$ and gradually increase it in every phase of the training.

In phases with $\alpha = 1$, no non-clustering loss is imposed, with potential disadvantages (see *No non-clustering loss* in Section 2.3). Similarly, in phases with $\alpha = 0$, no clustering loss is imposed, with potential disadvantages (see *No clustering loss* in Section 2.4).

## 2.6 CLUSTER UPDATES

Clustering methods can be broadly categorized into hierarchical and partitional (centroid-based) approaches (Jain et al., 1999). Hierarchical clustering combines methods which aim to build a hierarchy of clusters and data points. On the other hand, partitional (centroid-based) clustering groups methods which create cluster centers and use metric relations to assign each of the data points into the cluster with the most similar center.

In the context of deep learning for clustering, the two most dominant methods of each of these categories have been used. **Agglomerative clustering**, which is a hierarchical clustering method, has been used with deep learning (Yang et al., 2016b). The algorithm has been briefly discussed in Section 2.4. In addition, **k-means**, which falls into the category of centroid-based clustering, was extensively used (Xie et al., 2016; Yang et al., 2016a; Li et al., 2017; Hsu and Lin, 2017).

During the network training, cluster assignments and centers (if a centroid-based method is used) are updated. Updating cluster assignments can have one of the two following forms:

- **Jointly updated with the network model**: Cluster assignments are formulated as probabilities, therefore have continuous values between 0 and 1. In this case, they can be included as parameters of the network and optimized via back-propagation.

- **Alternatingly updated with the network model**: Clustering assignments are strict and updated in a different step than the one where the network model is updated. In this case, several scenarios are possible, dependent on two main factors:

  - **Number of iterations**: Number of iterations of the chosen clustering algorithm, that are executed at every cluster update step. For instance, in Xie et al. (2016), at each cluster update step, the algorithm runs until a fixed percentage of points change assignments between two consecutive iterations.
  - **Frequency of updates**: How often are cluster updates started. For instance in Yang et al. (2016b), for every $P$ network model update steps, one cluster updates step happens.

## 2.7 AFTER NETWORK TRAINING

Once the training converges, the network should have learned a mapping from the input space to a more clustering-friendly space with respect to the dataset it was trained on. In other words, if the training was performed on digit images of $N \times N$ pixel size, the network should be able to map a set of $N \times N$ images to a space where clustering is easier. With such a mapping, it makes sense to run a clustering algorithm on a desired dataset. However, the majority of the presented methods performs clustering during the training and obtain their clustering results from their last training iteration. Therefore the following are reasons for re-running the clustering after the training is done:

- **Clustering a similar dataset**: The general and the most trivial case is to reuse the learned features representation mapping on another dataset which is similar to the one that has been used but has different data.
- **Obtaining better results**: Under certain circumstances, it is possible that the results of clustering after the training are better than the ones obtained during the learning procedure. For instance, in Yang et al. (2016b), such a behavior is reported. One possible reason for this to happen is that the cluster update step during the training doesn't go all the way till the end (see *Number of iterations* in Section 2.6) meaning that older steps used older representations that might be worse. Therefore, some of the cluster merging steps (agglomerative clustering) were performed on a less optimal feature representation, which is why clustering after the training performed better.

## 3 RELATED METHODS

Clustering has been extensively studied and researched. Its application with deep neural networks has gained additional interest in the last few years, due to the success of supervised deep learning. However, in most cases, clustering is handled in an unsupervised fashion, making its application with deep learning less trivial and requiring more modeling effort and theoretical analysis. Therefore, several approaches have been presented over the last years, trying to use the representational power of DNNs for preprocessing clustering inputs. Each of these approaches used different network architectures, structures, loss functions and training methods in order to achieve their results and to improve the clustering quality. The following are some of the interesting methods that have been previously introduced.

### 3.1 DEEP EMBEDDED CLUSTERING (DEC)

DEC is one of the most promising approaches in the field. It is based on autoencoders as network architecture and initialization method, and uses $k$-means for clustering (Xie et al., 2016). As for training the neural network, the method first pretrains the model using a standard input reconstruction loss function. Secondly, the network's model is fine-tuned using the cluster assignment hardening loss and the clustering centers are updated. The clusters are iteratively refined by learning from their high confidence assignments with the help of the auxiliary target distribution. As a consequence, the method showed decent results and has later been used as a reference to compare new methods performances.

---

[1]Results from (Yang et al., 2016a) as the DEC paper did not publish results in NMI metric.

Table 1: Comparison of methods based on the taxonomy and quality of results. Quality numbers are from the respective original publications, except where otherwise noted.

| METHOD | ARCH | FEATURES FOR CLUSTERING | NON-CLUSTERING LOSS | CLUSTERING LOSS | COMBINING THE LOSS TERMS | CLUSTERING ALGORITHM | NMI MNIST | ACC MNIST | NMI COIL20 | ACC COIL20 |
|---|---|---|---|---|---|---|---|---|---|---|
| DEC (Xie et al., 2016) | MLP | Encoder output | Autoencoder reconstruction loss | Cluster assignment hardening | Pretraining and fine tuning | k-Means | 0.800[1] | 0.843 | - | - |
| DCN (Yang et al., 2016a) | MLP | Encoder output | Autoencoder reconstruction loss | k-Means loss | Alternating between joint training and cluster updates | k-Means | 0.810 | 0.830 | - | - |
| DEN (Huang et al., 2014) | MLP | Encoder output | Autoencoder reconstruction loss | - Group sparsity - Locality-preserving | Joint training | k-Means | - | - | 0.870 | 0.724 |
| DEPICT (Dizaji et al., 2017) | CNN | Encoder output | Autoencoder reconstruction loss | - Cluster assignment hardening - Balanced-assignment | Joint training | k-Means | 0.916 | 0.965 | - | - |
| DBC (Li et al., 2017) | CNN | Encoder output | Autoencoder reconstruction loss | Cluster assignment hardening | Pretraining and fine tuning | k-Means | 0.917 | 0.964 | 0.895 | 0.793 |
| JULE (Yang et al., 2016b) | CNN | CNN output | - | Agglomerative loss | - | Agglomerative clustering | 0.915 | - | 1 | - |
| CCNN (Hsu and Lin, 2017) | CNN | Internal CNN layer | - | Cluster classification loss | - | k-Means | 0.876 | - | - | - |
| Neural Clustering (Saito and Tan, 2017) | MLP | Concatenation of encoder layers output | Autoencoder reconstruction loss | - | - | kNN | - | 0.966 | - | - |
| Proposed | CNN | Encoder output | Autoencoder reconstruction loss | Cluster assignment hardening | Pretraining followed by joint training | k-Means | 0.923 | 0.961 | 0.848 | 0.762 |

## 3.2 DEEP CLUSTERING NETWORK (DCN)

DCN is another autoencoder-based method that uses $k$-means for clustering (Yang et al., 2016a). Similar to DEC, in the first phase, the network is pretrained using the autoencoder reconstruction loss. However, the second phase is different. In contrast to DEC, the network is jointly trained using a mathematical combination of the autoencoder reconstruction loss and the $k$-means clustering loss function. Thus, due to the fact that strict cluster assignments were used during the training (instead of probabilities such as in DEC) the method required an alternation process between the network training and the cluster updates. The method performed well and even led to better results than DEC on the MNIST dataset.

## 3.3 DISCRIMINATIVELY BOOSTED CLUSTERING (DBC)

With respect to the presented taxonomy, the approach in DBC (Li et al., 2017) is almost identical to DEC except for using convolutional autoencoders. Namely, it also uses $k$-means for clustering and the same training method: pretraining with autoencoder reconstruction loss and fine tuning using the cluster assignment hardening loss. Additionally, the same advantages and disadvantages are shared by both methods. Thus, due to the fact that DBC uses convolutional layers, it outperformed DEC's clustering quality on image datasets which was obviously expected.

## 3.4 JOINT UNSUPERVISED LEARNING OF DEEP REPRESENTATIONS AND IMAGE CLUSTERS (JULE)

JULE uses a convolutional neural network for representation learning. For clustering, a hierarchical approach is used, specifically, the agglomerative clustering method is employed. Concerning the training, the method only uses a clustering loss, specifically, the agglomerative loss. Additionally, the method has a period hyper-parameter, by which the training behavior is altered. Namely, this hyper-parameter specifies the number of model updates that should be applied before the clustering algorithm executes a clustering iteration, for instance, ten learning sessions followed by fusing two clusters into one. In experiments, the method showed great results, for example on MNIST, it performed better than all the other methods. However, the disadvantages of the lack of any non-clustering loss (see *No non-clustering loss* in Section 2.3) may be particularly pronounced, at least in theory (Yang et al., 2016a).

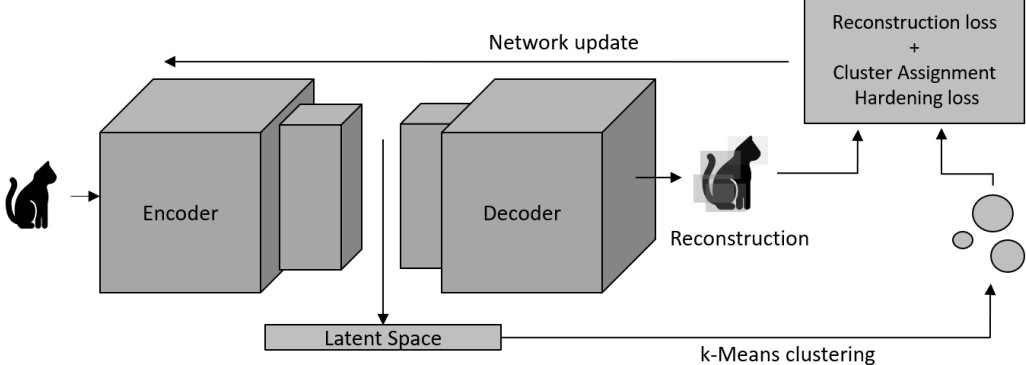

Figure 1: Proposed Method. We use a fully convolutional autoencoder, with reconstruction and cluster hardening loss, discussed in section 2.3 and 2.4 respectively, which results in cluster friendly feature space without the risk of collapsing to degenerate solutions

### 3.5 CLUSTERING CNN (CCNN)

CCNN uses a clustering CNN (Hsu and Lin, 2017) to achieve joint clustering and representation learning. One of the internal layers of the CCNN forms the feature space. At the same time, the CCNN's softmax layer predicts the cluster labels. Initially, features from k random images from the dataset are used to initialize the cluster centers. k-Means is performed on the features extracted from the input dataset to get corresponding cluster labels. Based on the assigned labels, and the labels predicted by the softmax layer, the CCNN parameters can be updated using the clustering classification loss discussed in section 2.4. The extracted features of the minibatch are then further used to update the corresponding cluster centroids.

### 3.6 OTHER METHODS

Besides the described methods, multiple attempts have been made in the field of clustering with deep learning. An interesting work is by Saito and Tan (2017) where a standard autoencoder was used without additional clustering loss functions. However, the outputs of several layers of the network beside the last one are used as the final feature representation. This layer concatenation led to superior results even when compared with methods which included a clustering-specific loss function. Moreover, in Huang et al. (2014), joint training was performed with a combination of an autoencoder reconstruction loss, a locality-preserving loss, and a group sparsity loss. Another work is by Dizaji et al. (2017), it is very similar to DEC, except for adding an additional term to the clustering loss which is a balanced assignments loss. By this addition, they alleviate the danger of obtaining degenerate solutions, but introduce again the need for alternating between network training and clustering updates. In addition to the mentioned methods, multiple others exist (Premachandran and Yuille, 2016; Harchaoui et al., 2017; Zheng et al., 2016; Chen et al., 2017; Lukic et al., 2016; Wang et al., 2016; Chen, 2015).

Rather than directly using a neural network to extract high-level features of samples, infinite ensemble clustering (Liu et al., 2016) uses neural networks to generate infinite ensemble partitions and to fuse them into a consensus partition to obtain the final clustering.

## 4 PROPOSED METHOD

After identifying a taxonomy of clustering with deep learning (Section 2) and comparing methods in the field based on it (Table 1), creating new improved methods became more straightforward. For instance, by looking at Table 1, one could notice that some combinations of method properties could lead to new methods. In some cases, such combinations could also surpass the limitations of the previous approaches and lead to better results. This procedure was followed during this work.

Namely, we picked an interesting combination of taxonomy features and came up with a new method (Fig. 1).

Our method uses a convolutional architecture, since our target clustering datasets are image datasets. Additionally, the network training has two phases. The first one is pretraining with an autoencoder reconstruction loss. In the second phase, the autoencoder loss and the cluster assignment hardening are jointly optimized. This second phase is different from DEC and DBC, which only use the cluster assignment hardening loss at this level. Omitting the reconstruction loss during one phase of the network training could lead to worse representations and solutions (see *No non-clustering loss* in Section 2.3). Therefore, combining the reconstruction loss with the cluster assignment hardening loss makes a lot more sense. This phase is also different from DCN, which has the joint training property, but uses the $k$-means loss. The $k$-means loss forces to alternate between joint training and clustering updates due to the hard cluster assignments. Using the cluster assignment hardening loss, this alternation procedure is no longer needed in our approach since this loss uses soft assignments which can be jointly updated with the network updates. Once both training phases are done, the network should be able to map its input into a more clustering-friendly space. Based on this assumption, we use the output of the network as the input to the $k$-means method which produces the final clustering results.

## 5 EXPERIMENTAL RESULTS

In this section we evaluate our model on real-world data and compare the results against the methods previously discussed in section 3.

**Validation Metrics**    For evaluation, we use the clustering accuracy (ACC) and normalized mutual information (NMI) metrics (Strehl and Ghosh, 2002; Vinh et al., 2010; Cai et al., 2011). These metrics lie in the range $[0, 1]$, with 1 being the perfect clustering, and 0 being the worst.

**Experimental Setup**    Training the network involved trying out several architectures and network sizes. In addition, it required tuning the learning hyper-parameters, such as the learning rate, initialization parameters, mini-batch size and others. In particular, we use a learning rate of $0.01$ with a momentum of $0.9$, in addition to batch normalization (Ioffe and Szegedy, 2015) and L2 regularization. The presented results are the best obtained ones during the experimentation phase.

**Datasets**    The experiments were performed on several publicly available datasets:

- **MNIST**: Consists of 70000 images of hand-written digits of $28 \times 28$ pixel size. The digits are centered and size is normalized (LeCun, 1998).
- **COIL20**: Contains 1440, $32 \times 32$ gray scale images of 20 objects (72 images per object). The images of each object were taken 5 degrees apart (Nene et al., 1996).

**Performance**    Table 1 shows the clustering performance in terms of accuracy and NMI for various clustering DNN approaches. The results for all the methods are borrowed from their respective publications. From the table it can be seen the proposed algorithm performs comparable, if not better than a lot of state of the art approaches.

Figure 2 and 3 show the clustering spaces at different stages of training the proposed network, with true cluster labels shown using different colors. The clustering spaces are 120-dimensional and 320-dimensional for MNIST and COIL20, respectively. It can be seen from the visualizations that the proposed method results in much more clustering-friendly spaces than the original image space and the autoencoder space.

## 6 CONCLUSION

In this work, we present a taxonomy for clustering with deep learning, identifying the general framework, and discussing different building blocks and possible options. In addition, a summary of methods in the field and their specific use of the taxonomy is presented alongside a general comparison

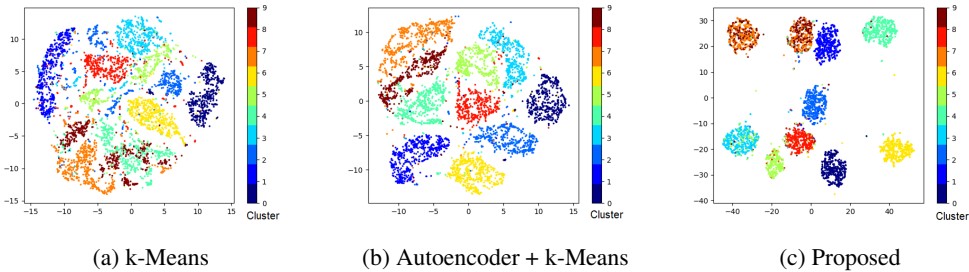

|(a) k-Means|(b) Autoencoder + k-Means|(c) Proposed|

Figure 2: $t$-SNE visualizations for clustering on MNIST dataset in (a) Original pixel space, (b) Autoencoder hidden layer space and (c) Autoencoder hidden layer space with the proposed method.

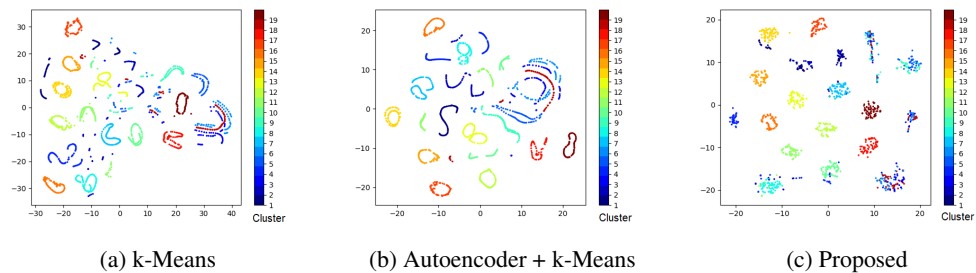

|(a) k-Means|(b) Autoencoder + k-Means|(c) Proposed|

Figure 3: $t$-SNE visualizations for clustering on COIL20 dataset in (a) Original pixel space, (b) Autoencoder hidden layer space and (c) Autoencoder hidden layer space with the proposed method.

of many of these methods. Using this taxonomy and the summary of previous methods, generating new methods is clearer and easier and can be done by creating new combinations of the taxonomy's building blocks. Moreover, we present a new method to the field, which is based on such a new combination. Our method overcomes the limitations of several previous ones, approaches state-of-the-art performance and performs better in some cases.

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
