# OpenReview forum: "Clustering with Deep Learning: Taxonomy and New Methods"
_ICLR.cc/2018/Conference — Reject_

### Official Review · AnonReviewer3 · 2017-11-23
**A paper combining kind of reviews and a not-so-useful clustering method**

**Rating:** 2
**Confidence:** 5

**Review:**

This paper presents some reviews on clustering methods with deep learning. Based on the review taxonomy, the authors presents a mixed objective which aims for bretter clustering performance. The proposed method is then tested on two image data sets.

The claimed main contribution of the paper is the taxonomy. There are no new things in such kind of reviews. The taxonomy gives no scientific axioms. Therefore the impact or actual contribution to the ICLR community is very limited.

The proposed clustering method is problematic. It is hard to set the paramter alpha. The experimental results are also disappointing. For example, the COIL20 accuracy is only 0.762, much worse than the state of the art. Moreover, results on only two image data sets are not sufficient for convincing.

---

### Official Review · AnonReviewer2 · 2017-11-25
**In this paper the authors give a nice review of clustering methods with deep learning and a systematic taxonomy for existing methods. However, the proposed new method is lack of novelty. The experiments part also need to be improved.**

**Rating:** 3
**Confidence:** 5

**Review:**

In this paper the authors give a nice review of clustering methods with deep learning and a systematic taxonomy for existing methods. Finally, the authors propose a new method by using one unexplored combination of taxonomy features.

The paper is well-written and easy to follow. The proposed combination is straightforward, but lack of novelty. From table 1, it seems that the only differences between the proposed method and DEPICK is whether the method uses balanced assignment and pretraining. I am not convinced that these changes will lead to a significant difference. The performance of the proposed method and DEPICK are also similar in table 1.

In addition, the experiments section is not comprehensive enough as well. the author only tested on two datasets. More datasets should be tested for evaluation. In addition, It seems that nearly all the experiments results from comparison methods are borrowed from the original publications. The authors should finish the experiments on comparison methods and fill the entries in Table 1.

In summary, the proposed method is lack of novelty compare to existing methods. The survey part is nice, however extensive experiments should be conducted by running existing methods on different datasets and analyzing the pros and cons of the methods and their application scenarios. Therefore, I think the paper cannot be accepted at this stage.

---

### Official Review · AnonReviewer1 · 2017-11-28
**A short survey and go at clustering with Deep Learning**

**Rating:** 3
**Confidence:** 4

**Review:**

The paper is mostly a survey about clustering methods with neural networks.

Section 2 presents a taxonomy for the different neural network clustering methods. A rich lists of the possible components of the neural network-based clustering methods are given, that include the different neural network architectures, feature to use for clustering, loss functions used and more. In Section 3, a few methods from the literature are classified according to the proposed taxonomy. Furthermore, in Section 4 a new method is proposed, that is to combine the best parts of the already existing models in the literature. Unfortunately, the experiments is Section 5 reveal that the proposed method yields results that are at most comparable with the existing methods.

The paper is written well and provides good insights (mostly taxonomy) on the existing methods for neural network-based clustering. However, the paper lacks novel content. The novel content of the paper sums up to the proposed method, that is composed of building blocks of existing models, and fails to impress in experimental results. It could be that this paper belongs to another venue that is more appropriate for survey papers.  Also, it overall rather appears short.

---

### Public Comment · (anonymous) · 2017-11-11
**Request for citation**

I believe that you should also cite “Learning Discrete Representations via Information Maximizing Self-Augmented Training” (ICML 2017) http://proceedings.mlr.press/v70/hu17b.html.
This paper is closely related to your work and is also about unsupervised clustering using deep neural networks.
As far as I know, the proposed method, IMSAT, is the current state-of-the-art method in deep clustering (November 2017). Could you compare your results against their result?

---

> ### Author Response · Authors · 2017-11-23
> **Re: Request for citation**
>
> Thank you for the hint. We checked the paper and it is definitely related to our work. We will make sure to include it in the next versions and compare its results. In addition, we'll try to add the losses introduced in this paper to our taxonomy.

---

### Decision · Program_Chairs · 2018-01-29
**ICLR 2018 Conference Acceptance Decision**

**Decision:**

Reject

**Comment:**

 + The paper offers a well-written survey and taxonomy of deep-learning approaches to clustering.
 - The novel proposed combination is not very original, and unconvincingly supported by experiments.
 => As the primary contribution of this paper seems to be its well-done survey, ICLR is probably not the appropriate venue for it.